# Cyber Dating Violence: How Is It Perceived in Early Adolescence?

**DOI:** 10.3390/bs14111074

**Published:** 2024-11-11

**Authors:** Iratxe Redondo, Naiara Ozamiz-Etxebarria, Joana Jaureguizar, Maria Dosil-Santamaria

**Affiliations:** 1Department of Developmental and Educational Psychology, Bilbao Faculty of Education, University of the Basque Country, 48940 Leioa, Spain; iratxe.redondo@ehu.eus (I.R.); joana.jauregizar@ehu.eus (J.J.); 2Department of Educational Sciences, Bilbao Faculty of Education, University of the Basque Country, 48940 Leioa, Spain; maria.dosil@ehu.eus

**Keywords:** cyber dating violence, intimate relationships, adolescence, aggression, cyber control

## Abstract

Background: Reports on cyber dating violence in adolescent populations vary significantly depending on whether the focus is on directly aggressive behaviours or behaviours designed to control one’s partner. In contrast to direct aggression, which is often clearly identified by adolescents, there is a greater degree of ignorance, and even a certain degree of normalisation, of controlling behaviours. Such behaviours may include, for example, insisting on knowing the whereabouts of a partner at all times or sharing social media passwords. This study aims to explore adolescent perceptions of cyber dating violence and to identify the differential characteristics of cyber-violent relationships using the Iramuteq software program for text analysis. Methods: Participants were 466 second- and third-year secondary school students. Data were collected through surveys, and responses were analysed using the Iramuteq program. This software tool enabled the identification of common terms and themes linked to cyber dating violence, as perceived by participating adolescents. Results: The analyses revealed the repeated appearance of terms associated with violent behaviours, online media, toxic relationships, and victim coercion. However, there was a notable lack of recognition of controlling behaviours as a manifestation of cyber dating violence. Adolescents frequently normalised behaviours such as insisting on knowing a partner’s whereabouts at all times or sharing social media passwords. Conclusions: The findings suggest a significant gap in adolescents’ understanding of what constitutes cyber dating violence. In contrast to direct aggression, which is easily identified, controlling behaviours are often normalised, indicating a need for educational and preventive measures to address this issue. By improving adolescents’ understanding of controlling behaviours as a form of cyber dating violence, preventive efforts can be more effectively tailored to address and mitigate this problem. To prevent the normalisation of certain behaviours indicative of cyber dating violence, early education is recommended in areas such as healthy relationships, communication skills, respect for privacy, and recognition of signs of excessive control.

## 1. Introduction

New technological advances and the cultural changes that accompany them have profoundly modified how we communicate, relate, and interact with each other [1]. Many of our relationships are now developed over digital platforms, which facilitates instant connection but can also generate dynamics of dependency and constant monitoring. Although the benefits of these technological advances are undeniable, such as instant access to information and easy ways to connect with others, they have also facilitated the emergence of new ways to exercise harassment, control, and abuse [2], rendering young adults and adolescents more vulnerable to intrusion and control in all social fields, including intimate partner relationships [3,4], in which this problem is often referred to as cyber dating violence or abuse.

### 1.1. Definition and Characteristics of Cyber Dating Violence

Cyber dating violence or abuse is a relatively new phenomenon in the scientific literature that can be defined as a set of repeated behaviours aimed at controlling, undermining, or harming one’s intimate partner or ex-partner [1,5]. In contrast to offline dating violence, cyber dating violence is characterised by the fact that it can be carried out quickly, easily, publicly, and at any place and time, even after the relationship has ended [6,7,8]. Thus, it has the potential to accentuate the experience of victimisation due to the public and ongoing nature of this type of aggression, as digital messages are permanent and can be easily shared [7,8].

As for the psychological factor related to this type of violence, several authors have found that cyber dating violence among adolescents is associated with greater depression and anxiety among victims. It also creates greater uncertainty about relationships, more antisocial behaviour, higher levels of hostility, and even higher levels of perceived stress than those caused by traditional abuse [9]. Moreover, another peculiarity of cyber dating violence is that, given that aggressions can occur online, at any time and place, the victim’s reactions may not always be clearly evident to the aggressor, which may lead to minimising the consequences of one’s actions [10]. This could also give the aggressor a sense of immunity rooted in anonymity [11].

Given that cyber dating violence refers to violence in dating relationships among adolescents, it is important to bear in mind that adolescence is a transitional life stage characterised by major physical, cognitive, and socioemotional changes [12], in which people often embark on their first romantic experiences, which are not always based on healthy relationship models [13]. Indeed, research in this field shows higher prevalence rates of both ‘offline’ and online intimate partner violence in adolescent relationships than among adults and young people [14,15]. It is therefore necessary to work with this population from a preventive perspective, in order to enable adolescents to develop the strategies they need to correctly identify and cope with cyber dating violence [16].

### 1.2. Perceptions and Normalisation of Cyber Dating Violence Among Adolescents

Previous research suggests that, among adolescents, the assessment of a behaviours as a possible instance of cyber dating violence varies depending on whether it is an example of direct aggression against the victim or is aimed rather at controlling their actions [5]. In the online environment, the first group of behaviours would include (among others) insults or other negative comments made in posts, emotional manipulation related to the content to be shared, and disclosure of private information [6,17,18]. These types of behaviour are often clearly identified by adolescents when asked about how they understand the term cyber dating violence [19,20].

However, adolescents do not always view behaviours aimed at controlling one’s partner or ex-partner, such as frequently visiting their social media profiles, monitoring their social media activity, or misusing their passwords [16,17], as something negative, and may even normalise them. For example, in a report published by Girlguiding [21], the authors found that 39% of the adolescent girls aged 11 to 17 in their sample believed that it was normal for their partner to tell them where they were at all times, and 22% thought that checking their mobile phone would be OK. In other words, certain behaviours indicative of cyber dating violence seem to have been normalised, either because adolescents consider them less serious or more acceptable than offline violence, or because they view them as relatively common. In their study involving young adults with a mean age of 22.72 years, Borrajo et al. [17] differentiated between direct aggression and control behaviours, finding that 10.6% of their participants admitted to having committed direct cyber aggression against their partner, and 82% admitted to having engaged in cyber control behaviours. Moreover, the prevalence of victimisation ranged from 14% (direct cyber aggression) to 75% (cyber control).

More recently, Cava et al. [22] found that almost half of the Spanish adolescents in their sample (44.1%) had never behaved in a cyber-controlling manner towards their partners, and more than a tenth (11.7%) had done so frequently. Similarly, in a qualitative study with adolescents aged 13–16, Stornad [23] found that many of the features of new technologies, and online relationships were perceived by many participants as activating cyber violence by facilitating controlling behaviours, anonymously and instantaneously.

### 1.3. Aim of the Present Study

Researchers in this field are therefore confronted with a complex phenomenon that adolescents do not always identify correctly, but which has negative consequences for them. The aim of the present study is therefore to improve our understanding of how adolescents perceive cyber dating violence. To this end, a qualitative methodology was used, and the idea was to gain insight into adolescents’ views on the basis of their own words and constructions, thereby overcoming the limitations inherent to data gathered through questionnaires, in which response options are pre-established. This will, in turn, provide a framework on which to base future preventive educational strategies specifically targeted at this type of violence and this developmental stage.

The present study is therefore significant because it explores how adolescents perceive cyber dating violence, a unique phenomenon in the digital age that can be easily normalised. By exploring these perceptions from a qualitative perspective, our findings will enable a deeper understanding of how young people view and experience these behaviours, providing insight that is essential for developing targeted preventive interventions and educational programs.

## 2. Materials and Methods

The sample group used in the present study comprised 459 students aged between 12 and 16 years (M = 13.73, SD = 0.93), 47.5% (n = 218) of whom were female, and 52.5% (n = 241) were male. Just over half (50.10%, n = 232) were in year 2 of compulsory secondary education, and 49.90% (n = 229) were in year 3. Additionally, 60.30% (n = 277) attended a semi-private subsidised school, and 39.70% (n = 182) a public school, both located in the province of Vizcaya (in the north of Spain). Overall, 402 (87.65%) were heterosexual, 44 (9.6%) bisexual, and 13 (2.8%) homosexual. In terms of relationships, 270 (60.30%) had never had a partner, 129 (28.10%) had had a partner in the past, and 53 (11.50%) currently had a partner. Among those who had previously had a partner, most had had one (75, 16.30%), two (39, 8.5%), or three (19, 4.1%) partners. Finally, the most frequent ages at which participants had their first partner were 13 (37, 8.10%) and 14 (32, 7%).

Before the study was launched, it was approved by the ethics committee of the University of the Basque Country (CEISH-UPV/EHU, M10/2018/2018). In order to ensure that school ownership was accurately represented, different semi-private and public schools in Vizcaya were contacted to explain the characteristics of the study. The schools that agreed to collaborate (2 semi-private and 1 public) provided students’ parents with information about the study and those who were willing to let their child(ren) take part signed an authorisation form. The questionnaires were administered in February 2022, collectively and during school hours. Participants were given a sheet of paper containing some general questions about their age, gender, school year, type of school, sexual orientation, and past and present romantic relationships. After these questions, the sheet included six blank spaces with the following instructions: ‘Please write down the first 3 words that come to mind when you hear the term cyber dating violence’. Next, to elicit a more detailed response, the following instruction was issued: ‘Now, explain in more detail what you mean by the first/second/third word you wrote down’.

In order to carry out a lexical analysis of the answers given, the Reinert method was used in conjunction with the Iramuteq software 0.8 alpha 7 version. Reinert [24,25,26,27] demonstrated that all discourses (regardless of the specific syntactic construction used) are made up of a set of words that can be taken as units of meaning.

The software functions as follows: Initially, the program classifies ‘whole words’ into nouns, verbs, adjectives, and adverbs. Next, the initial textual corpus is divided into text segments that are approximately one or two sentences long (40 words) [28,29]. The corpus is analysed for the presence of whole words in these segments. A contingency table is then constructed with the segments and reduced forms [25].

The software then runs a top-down hierarchical cluster analysis on this contingency table, extracting what is referred to as ‘classes’, i.e., sets of words that appear together and are clearly differentiated from other classes. Specifically, the program identifies the words and text segments with the highest chi-square values, i.e., those that best identify each class or idea repeatedly mentioned by participants. Finally, to interpret the results, it is advisable to assign a label or title to each class. The robustness of the Reinert method is based on the fact that the operations it performs are statistical, transparent, and reproducible right up to the final interpretation phase [30].

In the present study, after entering the raw data into the Iramuteq software, the most significant vocabulary items in each class were selected in accordance with the criteria established by Camargo and Bousfield [31]: (1) an expected word value greater than 3 and (2) the chi-square association test, contrasted with the class [χ2 ≥ 3.89 (*p* = 0.05); df = 1]. Based on the results obtained from the program, the text segments with the highest chi-square values in each class were recorded. Finally, labels or titles were assigned to each class. Labels are assigned by analysing the most significant words and text segments in each class. Using this information, the researcher tries to find a label that reflects what the elements in the class have in common, as part of a sense-making exercise. In our case, first of all, two of the researchers in the team labelled the classes independently and then discussed them in order to reach a consensus. In those cases in which there was no clear agreement, a third team member was brought in to help make the final decision, with the approval of the previous two.

## 3. Results

In order to carry out a more complex analysis, independent variables were entered into the program, categorised as follows: gender (male, female, non-binary); having had or not having had a partner (never, not currently but I did in the past, currently yes); number of partners so far (none, one, more than one); and age (12 years, 13 years, 14 years, 15 years, and 16 years). The association with these independent variables was interpreted taking into account a χ2 value of ≥6.63 (*p* = 0.01).

The complete textual corpus comprised 15,026 words, of which 1784 were single words. The top-down hierarchical analysis yielded a total of 529 ECUs (Elementary Contextual Units), which were grouped into six classes. The graph below (Figure 1) shows the classes and their significant words, as well as statistically significant differences (where appropriate) in accordance with the aforementioned variables.

The hierarchical grouping dendrogram identified six classes grouped into larger clusters. The first cluster, called ‘Characteristics of cyber dating violence (general)’, includes classes 4, 3, and 5, and refers to how participants understand cyber dating violence from a general point of view.

Class 4, with a weight of 20.4%, refers to aspects such as mistreating or hurting someone, either physically or verbally, which is why it was labelled ‘violence’. Significant differences (*p* < 0.001) were observed in accordance with gender (female χ^2^ = 6.87, *p* = 0.009). Some text segments characteristic of this class were as follows: ‘Mistreatment, when a person is not treated as they should be treated’ (female, no partner currently, one past partner, χ^2^ = 190.40), and ‘Aggressive comments, a comment made with the aim of hurting someone; aggressiveness, action designed to cause harm or other things; mistreatment, hurting someone or treating them badly’ (male, no partner currently, no past partners, χ^2^ = 180.83).

Within the same cluster, class 3 emerged with a weight of 18.9%. This class is associated with the means used to perpetrate violence and alludes to the technology or electronic devices used when engaging in this type of violent behaviour. Some of the characteristic text segments were as follows: ‘Social media, refers to the medium in which a series of events occur (...), the Internet, refers to the medium we use to access the social media or other websites’ (female, no partner currently, no past partners, χ^2^ = 173.83); ‘Computer, the type of cyber violence I assume takes place over the Internet and using technology’ (male, no partner currently, no past partners, χ^2^ = 159.55).

Finally, with a weight of 19.2%, class 5 is associated with the victim’s feelings. This class includes allusions to feelings or experiences such as anxiety, depression, and/or fear that are generated as a result of being attacked and or treated in this way by one’s partner. Some of the most significant text segments were as follows: ‘Fear, a fairly repetitive feeling caused by the situation of cyber violence’ (female, no partner currently, one past partner, χ^2^ = 117.36), and ‘Fear, it scares me because they can do anything to you, since you are not right in front of them, they feel like they can say anything to you’ (female, no partner currently, no past partners, χ^2^ = 112.00). Another example would be ‘In this case, I refer to the mutual manipulation, either online or in person, that turns a couple that once seemed lovely into a real headache that, in most cases, you can’t escape from’ (male, no partner currently, no past partners, χ^2^ = 106.58).

The second cluster, labelled ‘Experiences of cyber violence’, includes classes 6, 2, and 1, the latter two referring specifically to the offender’s behaviour. This second cluster includes participants’ personal or specific experiences of cyber dating violence.

Class 6, with a weight of 16.4%, is associated with the characteristics of the relationship. The most frequently mentioned characteristic is toxicity, with words such as toxic, toxic, toxic, and toxic (the program does not distinguish between the feminine/masculine and plural/singular forms of these words in Spanish). Participants also mentioned a lack of trust and lack of respect and pointed out that such behaviour may be mutual, i.e., that both partners may enter into this dynamic. Here, significant differences were observed in accordance with two variables: number of partners to date (having had more than one partner, χ^2^ = 8.60, *p* = 0.004) and gender (female, χ^2^ = 8.0, *p* = 0.004). Some of the most significant discursive fragments were as follows: ‘Toxic, relationships can become toxic and one partner can manipulate the other’ (female, no partner currently, more than one past partner, χ^2^ = 364.04); ‘Toxic, because if your partner hits you or you hit each other, you have a toxic relationship’ (female, no partner currently, no past partners, χ^2^ = 268.66); and ’A relationship becomes toxic when one of the partners takes advantage of the other, when apparently only one of the two benefits’ (male, no partner currently, one past partner, χ^2^ = 255.18).

Next is class 2, with a weight of 11.3%, which refers to the offender’s behaviour from the human point of view and includes aspects such as forcing someone to do things they do not want to do, annoying them, and making them feel bad. In other words, it refers to behaviours that directly affect the victim. Some of the most characteristic textual fragments in this class were ‘Forcing you to do something, forcing you to do things you don’t want to do. Checking things, he checks everything that is yours and doesn’t let you have any privacy’ (female, no partner currently, more than one past partner, χ^2^ = 344.81); ‘Victims feel obliged to do things they don’t want to do, don’t like or don’t feel like doing’ (female, no partner currently, no past partners, χ^2^ = 314.01); and ‘(...) when her partner doesn’t want to do something he mistreats her, he forces her to do something that she doesn’t want to do’ (male, no partner currently, no past partners, χ^2^ = 291.61).

Finally, there is class 1 (at more or less the same level as class 2), with a weight of 13.8%. This class is also related to the offender’s behaviour, although here, the focus is specifically on actions carried out over social media but those that, like the previous ones, have a direct effect on the victim. These behaviours include uploading intimate photos of the victim, leaking private information about them, or controlling their accounts, all without their consent. Some of the most significant text segments included the following: ‘Insulting a girl online, telling other people she is a xxxx and stuff like that, insulting her, uploading intimate pictures of her, uploading naked pictures and stuff like that and people seeing it, uploading private stuff about her, uploading conversations she wouldn’t want people to see’ (male, no partner currently, more than one past partner, χ^2^ = 356.90); ‘Not letting your partner go out, always checking their mobile phone or threatening to post private photos of them on the Internet’ (male, no partner currently, more than one past partner, χ^2^ = 346.76); ‘Sending an intimate photo to someone and it going viral (...), sending something private to your partner and him uploading it to the social media (...), sending him a video of you masturbating and him sharing it with his friends’ (male, no partner currently, no past partners, X^2^ = 332.07).

## 4. Discussion

In recent decades, ICT (information and communication technology) has become an important tool for information, entertainment, and communication among today’s adolescents, a generation that has been using the Internet and digital devices since childhood, with the consequent impact on their socialisation process, the construction of their identity, and the formation of their first relationships [20]. When used inappropriately, these technological advances have also given rise to new ways of perpetrating violence against partners or ex-partners [3].

The present study, which was carried out with students in years 2 and 3 of compulsory secondary education, provides information on how adolescent students perceive cyber dating violence and identifies the different ways in which it manifests in intimate partner relationships. The results obtained from an exploratory qualitative analysis of the responses given to open-ended questions on the research topic can be grouped into two general categories: general representation of the concept of cyber dating violence and specific characteristics of the relationships in which this type of violence occurs.

Firstly, regarding general perceptions of the concept of cyber dating violence, the adolescents in our sample group highlighted violent behaviour as being at the core of cyber violence, the digital media as the means by which it is exercised, and negative consequences for victims.

Participants clearly identified cyber dating violence as behaviours involving the mistreatment of another person, causing physical or verbal harm. It seems clear, therefore, that the findings are consistent with previous reports indicating that cyber dating violence is a type of intimate partner violence that includes any act of physical, emotional, or sexual violence occurring online or offline [14,20,32].

However, behaviours involving attempts to control one’s partner, such as repeatedly checking their publications on social media, usurping their identity, or insisting on constantly knowing where they are and who they are with, were not identified as instances of cyber dating violence, despite the fact that some previous studies have found that more than 80% of young people engage in such controlling behaviours in their intimate relationships [3,4,17]. Previous research indicates that some manifestations of cyber dating violence have become normalised and are even sometimes expected in adolescent dating relationships since, in the opinion of this population, they demonstrate trust and are a sign of professed mutual love [19,21].

Likewise, the definitions provided by the participants in our study clearly indicate that technology is used to exercise this type of violent behaviour. In other words, adolescents clearly and accurately identify that ICT can be used as an instrument for hurting a partner or ex-partner [33]. They also identify some of the feelings that victims may experience as a result of cyber dating violence, referring to reactions such as anxiety, depression, and fear, all of which are often mentioned in the scientific literature as consequences for the victim of this type of violence [9]. Indeed, bullying through social media and smartphones can have very serious consequences, including low self-esteem, psychological problems, anxiety, and depression [3].

Secondly, participants referred to the type of relationship in which cyber dating violence usually takes place, using terms linked to the characteristics of the relationship itself or the specific behaviours involved. When talking about the characteristics of the relationship, most respondents described relationships in which this type of violence occurs as toxic, alluding to the mutual dynamics that can emerge between the two partners. This perception is consistent with that reported by previous studies, which found that perpetration and victimisation are variables that are often related to each other (in both online and offline abuse), with victims frequently also being aggressors and aggressors tending to become victims [1,34]. However, the authors of these studies note that it is unclear whether perpetration occurred independently or as a reaction to victimisation. It is therefore important to explore the motivation, experience, and consequences of perpetration in more detail in order to better understand these dynamics [1].

Having had more previous partners and being a woman both seem to be associated with a more negative perception of cyber dating violence. This finding may be explained by the fact that having had more previous partners provides the individual in question with more experience in healthy relationships and fosters the perception of these behaviours as inappropriate. It also seems that being a woman may make respondents more aware of the ‘toxicity’ of these actions since women are usually the ones who suffer most from this type of violence. Indeed, according to figures published by UN Women, 73% of women in the world have suffered some type of cyber violence at some point in their lives.

However, these gender differences are not so clear in relation to controlling behaviours. In a recent study with young Basque university students, 42% of the total sample claimed to have suffered excessive control by their partner through social media or smartphones at least once, and 41.5% claimed to have perpetrated this type of violence towards their partner at least once. The study also found higher levels of control cyber violence victimisation among men than among women, although no gender differences were observed in perpetration [35]. Similarly, in another study with university students, Leisring and Giumetti [34] found no gender differences in either victimisation or the perpetration of ‘minor cyber abuse’, although men were more frequent perpetrators and victims of ‘major cyber abuse’. Contrary to these findings, a study conducted in the United States found that men reported more cyber victimisation than women [6]. Other studies with adolescent samples have found similar results, showing that although adolescence is a differentiated developmental stage, this phenomenon is nevertheless present in this age group. All this points to the need for prevention at an early age, even before young people begin to engage in their first romantic relationships [22,36,37].

In this second category referring to the specific characteristics of relationships in which cyber dating violence takes place, participants also highlighted the specific actions carried out by aggressors, mentioning (among others) ‘forcing another person to do things’, as well as behaviours such as uploading photographs and nude photographs and sharing private information. It is possible here to infer relationship patterns characterised by the superiority of one of the partners, in which power does not occur without violence. However, consistent with that discussed above, behaviours aimed at controlling one’s partner’s activity on social media are once again underrepresented or normalised [18,19,20].

### Limitations

The present study has some limitations that should be taken into account when generalising the results. First, working with a larger sample would have provided a broader understanding of adolescents’ perceptions of cyber violence. Future studies should therefore strive to recruit larger samples that are more representative of the adolescent population as a whole. It is also important to remember that the population studied included only students aged 12 to 16 years. It may be interesting to include 16-to-18-year-olds in future samples since those in this age group are still adolescents and might have more experience in dating relationships. Another limitation of the present study was that it did not take into account whether or not adolescents had experienced cyber dating violence in their previous relationships, something that should be redressed in future research. Finally, future studies may wish to combine data obtained through open-ended questions with those obtained from interviews, observations, and/or focus groups in order to gather more comprehensive qualitative information from participants.

## 5. Conclusions

The present study constitutes an exploratory approach to adolescents’ perceptions of the phenomenon of cyber dating violence, suggesting that, although this population seems to have adequate general knowledge of this type of violence and its manifestations, more prevention is required to avoid the normalisation of certain warning signs, such as the exchange of passwords and the monitoring and control of the other person’s social media activity, since said normalisation makes their detection more difficult. Given the relationship between cyber violence and other types of violence and abuse [2,16,34], as well as its well-documented adverse effects on the victim [9], it is vital to implement preventive interventions as early on as possible.

Preventive interventions that could be implemented include early education on healthy relationships in both educational and community settings and actions designed to foster communication and conflict resolution skills and promote respect for privacy in relationships. It would also be useful to train adolescents to recognise warning signs related to cyber dating violence, such as excessive control. Programs that address these issues could help prevent the normalisation of these behaviours [38].

## Figures and Tables

**Figure 1 behavsci-14-01074-f001:**
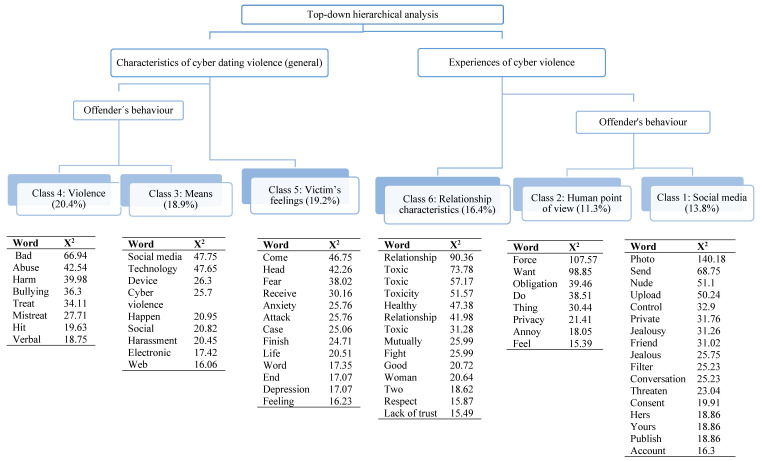
Hierarchical grouping dendrogram of the free-association exercise, showing the most frequent words and the words with the highest association indexes χ^2^(1), *p* < 0.001. Source: the authors.

## Data Availability

Data can be obtained by contacting the corresponding author.

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
