# Peer review of "Cyber Dating Violence: How Is It Perceived in Early Adolescence?"

_behavsci, 2024, doi:10.3390/bs14111074_

Round 1
Reviewer 1 Report (Previous Reviewer 2)
Comments and Suggestions for Authors
The authors made revisions to the manuscript, responded to the reviewers' main comments, added missing information in the methods, made the discussion clearer, and completed the limitations of the study.
Authors are requested to check for possible typos and omissions: line 91-92, 237, 301, 346 (missing reference).
Author Response
Good morning
Thank you very much for your reviews
Thanks to them, the article is much improved.
We have made all the suggested changes and an expert native translator has checked the English of the whole article.
Thanks to the two rounds of revisions we believe that this article has improved a lot scientifically.
Thank you
Below is my response to the review
REVIEWER 1
The authors made revisions to the manuscript, responded to the reviewers' main comments, added missing information in the methods, made the discussion clearer, and completed the limitations of the study.
Authors are requested to check for possible typos and omissions: line 91-92, 237, 301, 346 (missing reference).
Thank you very much. We have made the corrections.
Reviewer 2 Report (New Reviewer)
Comments and Suggestions for Authors
Title: Cyber dating violence: how do early adolescents perceive it?
General comment:
The authors have conducted a study in order to explore adolescents’ perceptions about cyber dating violence and its related characteristics. I acknowledge the importance of examining adolescents’ opinions and perceptions about this topic. Nevertheless, the manuscript could benefit from a clearer structure and from an English editing. Moreover, the study could be better justified.
Specific comments:
Abstract
· The abstract is well-structured and well summarizes the study. Nevertheless, it would benefit from a more detailed statement of the study importance and potential practical implications in terms of intervention.
· Line 16-17: I suggest rephrasing the sentence to allow a better understanding.
Introduction
In general, the Introduction could be better structured. For instance, the authors firstly talk about new technologies and the consequent changes in communication and interaction, then they provide prevalence data and briefly introduce benefits related to new technologies. Thereafter, authors define cyber dating violence and highlight its characteristics, then talk about psychological consequences and back to cyber dating violence characteristics…
The authors should aim to find a more strict and well-organized presentation of the background, a clearer line of reasoning that guides the reader through the theoretical background.
I would also suggest to stress more why this study is important and what it adds to the literature.
· Page 1, Line 37-38: the authors claim that new technologies have deeply changed the way adolescents communicate and interact. I suggest explaining in which ways the advent of new technologies has affected this aspect.
· Page 1, Line 39: the authors introduce some rates (i.e., adolescents’ use of Internet, percentage of adolescents who have a mobile phone) from “latest studies”. However, these recent studies are not mentioned. I suggest citing them. Moreover, I suggest to relocate this part, because here it seems to fragment the discourse.
· Page 1, Line 41: the authors talk about benefits related to new technologies. I suggest to add examples.
· Page 1, Line 44: “in which context”: I suggest to eliminate the word “context”.
· Page 2, Line 49: there’s a typo. Change “characterised” in “characterized”.
· Page 2, Line 54: I suggest rephrasing the sentence “As for the psychological factor related to this kind of violence” and to add a sentence introducing this section after talking about cyber dating violence characteristics. Indeed, I suggest to ensure a smoother transition between different sections: it could be helpful for the reader to follow the argument.
· Page 2, Line 58: the authors cite a review (Hinduja & Patchin, 2011). I suggest to cite more recent studies.
· Page 2, Line 61: there’s a typo. Please, change “minimise” in “minimized”.
· Page 2, Line 87: please rephrase the following sentence: “with some researchers even finding a normalisation effect”.
· Page 2, Line 94-95: the authors write “in adolescents with a mean age 94 of 22.72 years”. That is not a sample of adolescents (or, at least, of just adolescents). I suggest highlighting if the sample comprised also young adults (in that case please specify it) or find a study only focused on adolescents.
· Page 2, Line 95-96: please, rephrase the sentence.
· Page 2, Line 98: please, eliminate the comma after “75%”.
· Page 3, Line 99: “Cava et al.[23] that almost”. Please, complete the sentence.
· Page 3, Line 101: please, rephrase the sentence for a clearer communication.
· Page 3, Line 106-107-108: please, rephrase the sentence for a clearer communication.
Materials and Methods
The sample size of 459 students seems adequate and it also seems to be well balanced.
· Page 3, from line 117 to 127: please check the percentages. In some case “%” is missing when presenting data.
· Page 3, Line 137-138: I would summarize this part in broad categories. E.g., romantic relationship characteristics (i.e., sexual orientation, whether they had had a partner before or had a partner now, and if they had had a partner, how many partners they had had so far and at what age they had their first partner).
Results
· Page 4, Line 177: please modify “categorised” in “categorized”.
· Page 4, Line 183: when presenting numbers use “.” and not comma (e.g., 15,026).
· In Figure 1 there’s a typo: in the left part of the graph modify “offenders behaviour” in “offenders’ behaviour”. Also, regarding Figure 1, make sure to use British English, as in the text.
· Page 5, Line 202: “p <.01” but in the graph is reported “p <.001” under class 4.
· Page 6, Line 227: in the text the authors call the second cluster “Cyber violent relationships” but in Figure 1 it has been labelled as “Cyber violence experiences”.
· Page 6, Line 236-237: please, revise the sentence and verify typos.
Discussion
· Page 7, Line 301: please unify the word “previous”.
· Page 8, from line 322 to 325: I suggest to further explain this aspect to ensure a clearer understanding.
· Page 8, Line 332: the authors cite “UN Women”… Is it from a survey? Is it possible to find other data?
Conclusions
· Page 9, Line 377: please unify “the”
Comments on the Quality of English LanguageThe manuscript could benefit from an English editing. I also suggest to choose British or American English and stick with it throughout the text.
Author Response
Good morning
Thank you very much for your reviews
Thanks to them, the article is much improved.
We have made all the suggested changes and an expert native translator has checked the English of the whole article.
Thanks to the two rounds of revisions we believe that this article has improved a lot scientifically.
Thank you
Below are the responses to the reviews
REVIEWER 2
General comment:
The authors have conducted a study in order to explore adolescents’ perceptions about cyber dating violence and its related characteristics. I acknowledge the importance of examining adolescents’ opinions and perceptions about this topic. Nevertheless, the manuscript could benefit from a clearer structure and from an English editing. Moreover, the study could be better justified.
Specific comments:
Abstract
- The abstract is well-structured and well summarizes the study. Nevertheless, it would benefit from a more detailed statement of the study importance and potential practical implications in terms of intervention.
- Line 16-17: I suggest rephrasing the sentence to allow a better understanding.
Thank you, we have underlined the importance of the study and the possible practical implications in terms of intervention.
We have also rephrased the sentence to allow for a better understanding.
Introduction
In general, the Introduction could be better structured. For instance, the authors firstly talk about new technologies and the consequent changes in communication and interaction, then they provide prevalence data and briefly introduce benefits related to new technologies. Thereafter, authors define cyber dating violence and highlight its characteristics, then talk about psychological consequences and back to cyber dating violence characteristics…
The authors should aim to find a more strict and well-organized presentation of the background, a clearer line of reasoning that guides the reader through the theoretical background.
Thank you, we have emphasised why this study is important and what it contributes to the literature. We have also structured the introduction.
I would also suggest to stress more why this study is important and what it adds to the literature.
- Page 1, Line 37-38: the authors claim that new technologies have deeply changed the way adolescents communicate and interact. I suggest explaining in which ways the advent of new technologies has affected this aspect.
Thank you, we have explained
- Page 1, Line 39: the authors introduce some rates (i.e., adolescents’ use of Internet, percentage of adolescents who have a mobile phone) from “latest studies”. However, these recent studies are not mentioned. I suggest citing them. Moreover, I suggest to relocate this part, because here it seems to fragment the discourse.
Thank you, we have removed this sentence from here
- Page 1, Line 41: the authors talk about benefits related to new technologies. I suggest to add examples.
Thank you, we have explained
- Page 1, Line 44: “in which context”: I suggest to eliminate the word “context”.
thank you, we have removed it
- Page 2, Line 49: there’s a typo. Change “characterised” in “characterized”.
thank you, we have changed it
- Page 2, Line 54: I suggest rephrasing the sentence “As for the psychological factor related to this kind of violence” and to add a sentence introducing this section after talking about cyber dating violence characteristics. Indeed, I suggest to ensure a smoother transition between different sections: it could be helpful for the reader to follow the argument.
Thank you, we have reformulated the sentence and ensured a smoother transition.
- Page 2, Line 58: the authors cite a review (Hinduja & Patchin, 2011). I suggest to cite more recent studies.
Thank you, we have included a more up to date article
- Page 2, Line 61: there’s a typo. Please, change “minimise” in “minimized”.
thank you, we have corrected it
- Page 2, Line 87: please rephrase the following sentence: “with some researchers even finding a normalisation effect”.
thank you we have rephrased it
- Page 2, Line 94-95: the authors write “in adolescents with a mean age 94 of 22.72 years”. That is not a sample of adolescents (or, at least, of just adolescents). I suggest highlighting if the sample comprised also young adults (in that case please specify it) or find a study only focused on adolescents.
thank you, we have modified it
- Page 2, Line 95-96: please, rephrase the sentence.
thank you we have rephrased it
- Page 2, Line 98: please, eliminate the comma after “75%”.
thank you we have eliminated it
- Page 3, Line 99: “Cava et al.[23] that almost”. Please, complete the sentence.
thank you we have completed it
- Page 3, Line 101: please, rephrase the sentence for a clearer communication.
thank you we have rephrased it
- Page 3, Line 106-107-108: please, rephrase the sentence for a clearer communication.
thank you we have rephrased it
Materials and Methods
The sample size of 459 students seems adequate and it also seems to be well balanced.
- Page 3, from line 117 to 127: please check the percentages. In some case “%” is missing when presenting data.
Thank you very much we have put the percentages
- Page 3, Line 137-138: I would summarize this part in broad categories. E.g., romantic relationship characteristics (i.e., sexual orientation, whether they had had a partner before or had a partner now, and if they had had a partner, how many partners they had had so far and at what age they had their first partner).
Thank you very much, we have followed your recommendation
Results
- Page 4, Line 177: please modify “categorised” in “categorized”.
Ok, thank you
- Page 4, Line 183: when presenting numbers use “.” and not comma (e.g., 15,026).
Ok, thank you
- In Figure 1 there’s a typo: in the left part of the graph modify “offenders behaviour” in “offenders’ behaviour”. Also, regarding Figure 1, make sure to use British English, as in the text.
perfect, changed and improved the English of the figure
- Page 5, Line 202: “p <.01” but in the graph is reported “p <.001” under class 4.
perfect, changed
- Page 6, Line 227: in the text the authors call the second cluster “Cyber violent relationships” but in Figure 1 it has been labelled as “Cyber violence experiences”.
perfect, changed
- Page 6, Line 236-237: please, revise the sentence and verify typos.
perfect, changed
Discussion
- Page 7, Line 301: please unify the word “previous”.
thank you, we have unified it
- Page 8, from line 322 to 325: I suggest to further explain this aspect to ensure a clearer understanding.
- Page 8, Line 332: the authors cite “UN Women”… Is it from a survey? Is it possible to find other data?
Thank you, we have explained this sentence better
Conclusions
- Page 9, Line 377: please unify “the”
unified
Round 2
Reviewer 2 Report (New Reviewer)
Comments and Suggestions for Authors
General comment:
The authors have very much improved the quality of their article, which now appears to be well structured and exhaustively detailed. They also better specified the added value of the study in the literature on the topic.
Specific comments:
Introduction
· Page 2, line 148-149: I suggest to rephrase this sentence “which may in turn prompt them to minimise the consequences of their actions, or even give them a sense of immunity rooted in anonymity” in this way: “which may be led to minimise the consequences of one’s actions. This could also give the aggressor a sense of immunity rooted in anonymity”.
· Page 2, line 157: with “young people” are the authors referring to young adults? If so, please specify it.
· Page 2, from line 159 to 162: I suggest to eliminate from here the sentence introducing the aim of the study, because after this paragraph the authors are still reporting evidence from the literature on the theme. Also, the aim is well explained in the last paragraph of the Introduction section.
· Page 2, line 171: please, correct “behaviour” in “behaviours”.
· Page 2, line 176: please eliminate the comma after “[22]”.
Materials and Methods
· Page 3, line 324: “age,gender, school year , type of school”. Please, correct the commas: “age, gender, school year, type of school”.
Results
· Page 4, line 499: please, correct “having had or not had a partner” with “having had or not having had a partner”.
· Page 6, line 657: please, eliminate the comma after “although here”.
Discussion
· Page 7, line 742: please, add “the” before the word “victims”.
· Page 7, line 743: please, modify “behaviour” in “behaviours”.
· Page 7, line 745: please, modify “consistently with that reported previously” with “consistently with what reported previously”.
Conclusions
· Page 9, line 921: please, rephrase this sentence “[…] phenomenon of cyber dating violence, allowing us to conclude that, although […]” in this way “[…] phenomenon of cyber dating violence, suggesting that, although […]”.
Author Response
Dear Reviewer.
Thank you very much for reviewing the article.
Thanks to your detailed review we believe it has been improved.
Below are the revisions with the responses to them.
We really appreciate the work you have done.
Thank you
Kind regards
Introduction
Page 2, line 148-149: I suggest to rephrase this sentence “which may in turn prompt them to minimise the consequences of their actions, or even give them a sense of immunity rooted in anonymity” in this way: “which may be led to minimise the consequences of one’s actions. This could also give the aggressor a sense of immunity rooted in anonymity”.
Thank you very much, we have rephrased it
- Page 2, line 157: with “young people” are the authors referring to young adults? If so, please specify it.
Thank you very much we have changed it
- Page 2, from line 159 to 162: I suggest to eliminate from here the sentence introducing the aim of the study, because after this paragraph the authors are still reporting evidence from the literature on the theme. Also, the aim is well explained in the last paragraph of the Introduction section.
We have deleted it
- Page 2, line 171: please, correct “behaviour” in “behaviours”.
Thank you very much we have changed it
- Page 2, line 176: please eliminate the comma after “[22]”.
We have deleted it
Materials and Methods
- Page 3, line 324: “age,gender, school year , type of school”. Please, correct the commas: “age, gender, school year, type of school”.
Thank you, we have corrected it
Results
- Page 4, line 499: please, correct “having had or not had a partner” with “having had or not having had a partner”.
Thank you, we have corrected it
- Page 6, line 657: please, eliminate the comma after “although here”.
Eliminated
Discussion
- Page 7, line 742: please, add “the” before the word “victims”.
We have added “the”
- Page 7, line 743: please, modify “behaviour” in “behaviours”.
Ok thank you
- Page 7, line 745: please, modify “consistently with that reported previously” with “consistently with what reported previously”.
Ok, perfect
Conclusions
- Page 9, line 921: please, rephrase this sentence “[…] phenomenon of cyber dating violence, allowing us to conclude that, although […]” in this way “[…] phenomenon of cyber dating violence, suggesting that, although […]”.
Thank you we have rephrased it
This manuscript is a resubmission of an earlier submission. The following is a list of the peer review reports and author responses from that submission.
Round 1
Reviewer 1 Report
Comments and Suggestions for Authors
In general a nice article to read, but the instruction needs some restructuring, the recruitment procedure has to be elaborated, the ground on which some conclusions are made is unclear (and that’s a major flaw!) and the literature needs to be updated.
Abstract
The abstract is well written
Introduction
The introduction starts well, but then shifts from one topic to another. The transition between the paragraphs is not always well made. For example, make it clear on line 63 why the focus is on adolescents (e.g. move the short paragraph lines 71-73 up, e.g. move the short paragraph lines 92-96 up)
Check for language spelling
e.g., line 97: verb is missing
Please avoid “normative” language.
Line 59: replace “attacks” by another word
Line 104: remove “we”
Line 110: replace “us” by another word
Methods
Lines 113 and following: explain the recruitment procedure + when was the administration + ….: provide more details.
Lines 142 and following: explain how consensus is created, who is interpreting the findings, and how labels or titles are assigned to each class
Lines 148: explain how the value 3 is created/obtained
Lines 151 and following: explain how labels or titles are assigned to each class. Are these different that those from lines 142 and following. Explain this thoroughly
Results
Line 157: there are 9 non-binary persons. With a chisquare test, this will result in cells with less then 5 cases, making interpretation touchy. Fisher z analyses are more appropriate, or remove the non-binary persons and do the analyses again.
Line 209: given the number of words, I suggest to use p<.01 as significance level (instead p<.05)
Include consistent chisquare statistics and p levels
Conclusion
Lines 266-276: how can you conclude based on the data that there is a normalization. See figure 1. Please provide more clear evidence for this. It seems that the conclusions do not follow the data.
Lines 277-286: how can you conclude that adolescents clearly and accurately identify the means? Please elaborate thoroughly. It seems that the conclusions do not follow the data.
Lines 301-304: this is not true. See https://doi.org/10.1177/15248380221082087
Lines 322-324: how can you conclude this based on the data?
References
Update the reference list with more recent literature
Comments on the Quality of English Language
ok
Reviewer 2 Report
Comments and Suggestions for Authors
The work focuses on the urgent topic of adolescents' perception of cyber dating violence. Taking into account the role of digital interactions and age-psychological characteristics of adolescents, this topic seems very relevant. The study is qualitative in nature and seeks to understand perceptions of cyber dating violence among adolescents. The overview part reveals the main points of view on the problem and justifies the chosen research design. The sample of 466 adolescents seems to be sufficiently large for a qualitative study. The methods describe in detail the Iramuteq program used for text processing. The methods of data collection and data processing are consistent with the purpose of research. The results of the study are presented correctly and give a full understanding of the findings. In the discussion of the results, the authors of the study provide an interpretation of the findings and relate them to other empirical studies, as well as present the limitations of the results.
The work is carried out to a quite good level, following all the basic scientific requirements. Among the suggestions for improvement, it would be desirable to emphasize a few points.
1. Although the study is searching phenomenological in nature, aimed at identifying adolescents' perceptions, the paper nevertheless lacks hypotheses. For example, in the review section, the authors point out, with reference to other studies, that adolescents may be good at distinguishing direct violence, but poor at identifying a partner's controlling behavior as violence. On the basis of this and other aspects of the overview, it would be possible to formulate hypotheses.
2. It might be worth reflecting in the methods that in addition to answering open-ended questions, respondents answered questions about their experience of partnerships. These variables appear only in the results.
3. Was it clarified with the adolescents whether, as part of their relationship experience, they had experienced cyber dating violence and in what role? This may have been an important factor in shaping perceptions of the problem. If the question was not asked of respondents, it is worth reflecting this in limitations.
4. The age dimension in the perception of cyber dating violence may be significant. The data collected provide an opportunity to examine how younger and older adolescents evaluate cyber dating violence and to identify the absence or presence of differences.
5. A more structured presentation of the discussion, limitations and conclusions could be recommended as separate sections. Everything is reflected in the text, but within the discussion section.
Reviewer 3 Report
Comments and Suggestions for Authors
Generally speaking, I thought the paper explored an interesting topic, and provide some insightful findings. And this study used innovative methodology. But I have several concerns regarding the methodology and findings too. 1. The major concern was the participants. As the authors stated in the limitation part, the participants in the current study were around age 12 to 16, the second and third years of secondary schools. For adolescents in this age, they may be not quite familiar with dating/romantic relationships and many may not get involved in dating relationships. On the other hand, many early adolescents are highly involved in general peer interactions. Thus, the validity of participants' answers regarding the open-ended question "write down the first 3 words that come to mind when you hear the term cyber dating violence" may be questioned. Given the fact that the participants were early adolescents, and experiencing transition from childhood to adolescent years, some of the results (e.g., they did not report giving/asking passwords as manipulation or dating violence) may be not surprising. At least, I suggest the authors change the title of the paper to indicate this is research on early adolescents. 2. some of the basic information, such as how many (percentage) of participants had no experience of dating, etc, should be reported. 3. I am not familiar with the Iramuteq program and the textual analyses. But regarding Figure 1, I cannot generally tell the substantial differences between cluster 1 and cluster 2, especially regarding the cluster name "Characteristics of cyber dating violence" and "cyber violent relationships “. In another words, the authors shall provide more valid (theoretically-based valid) explanation/analyses regarding the clustering results (by Iramuteq program) of the textual corpus. 4. for the literature review, it would be better that the authors provided info on the age/developmental stage of the paricitipants in extant studies, for example, the studies by Girlguiding, Borrajo et al. etc. 5. in the discussion of the results, it would be better too that the authors take the developmental stages of participants into consideration, when discussing the current results. Resulst of understanding of cyber violence of university students and secondary school students shall show differences. The authors need to keep developmental differences in mind, at least in discussing the current findings.
